# The Effect of Acetylene Carbon Black (ACB) Loaded on Polyacrylonitrile (PAN) Nanofiber Membrane Electrolyte for DSSC Applications

**DOI:** 10.3390/mi14020394

**Published:** 2023-02-04

**Authors:** Herlin Pujiarti, Zahrotul Ayu Pangestu, Nabella Sholeha, Nasikhudin Nasikhudin, Markus Diantoro, Joko Utomo, Muhammad Safwan Abd Aziz

**Affiliations:** 1Department of Physics, Faculty of Mathematics and Natural Sciences, Universitas Negeri Malang, Jl. Semarang 5, Malang 65145, Indonesia; 2Centre of Advanced Materials for Renewable Energy (CAMRY), Universitas Negeri Malang, Jl. Semarang 5, Malang 65145, Indonesia; 3Faculty of Science, Universiti Teknologi Malaysia, 05-07 Level 5 Block T05 Laser Center, Skudai 81310, Johor, Malaysia

**Keywords:** DSSC, nanofiber membrane, efficiency, polyacrylonitrile, acetylene carbon black

## Abstract

Nanofiber membranes are starting to be used as an electrolyte storage medium because of their high porosity, which causes ionic conductivity, producing high energy. The ability of nanofiber membranes to absorb electrolytes proves their stability when used for a long time. In this study, the loading of acetylene carbon black (ACB) on polyacrylonitrile (PAN) is made by the electrospun method, which in turn is applied as an electrolyte medium in DSSC. Materials characterization was carried out through FTIR to determine the functional groups formed and SEM to observe morphology and diameter distribution. Furthermore, for DSSC performance, efficiency and EIS tests were carried out. The optimum nanofiber membrane was shown by esPACB1, with the highest efficiency reaching 1.92% with a porosity of 73.43%, nanofiber diameter of 172.9 ± 2.2 nm, an absorbance of 1850, and an electron lifetime of 0.003 ms.

## 1. Introduction

Dye-sensitized solar cell (DSSC) is an alternative to conventional solid-state devices that are cost-effective and environmentally friendly [1]. Its mechanism involves using dye to absorb and convert solar energy into electrical energy [2,3,4]. O’regan and Gratzel developed the first DSSC in 1991 with an efficiency of approximately 13% [5]. DSSC consists of a photoanode, an electrolyte, and a conductive substrate coated with a catalyst, such as Pt and C, as the counter electrode [6,7]. An electrolyte is a crucial component in dye-sensitive solar cells, which serves as a medium for charge transport between the photoanode and counter electrode. The electrolyte must have high ionic conductivity and good thermal, optical, chemical, and electrochemical stability over the long term. The higher the ionic conductivity of the electrolyte, the higher the cell performance. Achieving superior ionic conductivity is essential for any electrolyte to be applied in electrochemical devices [8]. Liquid electrolytes are widely developed due to their low viscosity, high ionic conductivity, simple fabrication, and high energy conversion efficiency [9]. Hilmy et al. stated that the ionic conductivity of a liquid electrolyte-based DSSC is approximately 9.6751 mS·cm^−1^ [10]. Wang et al. reported that the efficiency of a 10% ionic electrolyte-based DSSC was stable in full sunlight at 60 °C with a heat stress of 85 °C for 1000 h [11]. However, this device had several weaknesses, including solvent leakage, electrode corrosion and low chemical and thermal stability [12].

Presently, quasi-solid electrolytes are classified as good DSSC candidates due to their ability to produce optimum efficiency, high ionic conductivity, non-volatile electrolytes, and good contact with electrodes [13,14]. According to several preliminary studies, polymer-based gel electrolytes increase long-term stability [15]. Aziz et al. reported the efficiency of a polyvinyl alcohol (PVA) polymer electrolyte gel-based DSSC containing a mixture of iodide salts of 6.4% with the highest ionic conductivity of 12.48 mS·cm^−1^ [15]. To date, electrolyte gel has shown relatively low mechanical strength, limiting its application for large-scale production [16,17].

Nanometer-scale membranes are currently developed as electrolyte media from polymeric materials made using the electrospun method because they are cheap and easy to use to produce nanofibers from polymer solutions. Electrolytic membranes have a high surface-area-to-volume ratio, good mechanical performance [18], high ionic conductivity due to their high porosity, and exceptional electrolyte-absorbing ability [1]. Various kinds of polymers have been used for the synthesis of polymer membranes, namely poly (methyl methacrylate) (PMMA), poly (acrylonitrile) (PAN), and poly (vinylidene fluoride-hexafluoropropylene)/poly (vinyl alcohol) (PVDF-HFP/PVA) [15]. PAN is an environmentally friendly conductive polymer with good mechanical and thermal stability and electrical and optical properties [19]. Electrolyte membranes based on PAN/CoS nanocomposites reach an efficiency of 7.41%, which is higher than the electrospun PAN-based membrane value of 6.47% [1]. In addition, PVDF-g-PAN was also developed as an electrolyte membrane for DSSC and produced an efficiency of 2.08% [20].

Numerous studies have been conducted focusing on the possibility of combining polymer and additive nanoparticle composites that can significantly influence the properties of such material for the desired application. Furthermore, the incorporation of acetylene carbon black (ACB) onto PAN nanofiber composite as a polymer membrane electrolyte in DSSC has not been critically addressed elsewhere. ACB is believed to increase the active surface area, which affects its ionic conductivity and overall DSSC photovoltaic performance. ACB has a high surface-area-to-volume ratio and low crystallinity, which can reduce space and increase charge transfer ability and corrosion resistance to iodine [21]. This work demonstrated the PAN-ACB membrane composite by directly electrospinning nanofibers onto aluminum foil substrate. The effect of different ACB mass incorporated onto membrane composite towards the electrolyte’s absorbance ability and ionic conductivity is discussed in detail for its potential application in DSSC.

## 2. Materials and Methods

### 2.1. Synthesis of PAN Membranes and PAN/ACB Nanofiber Composite

Figure 1 shows synthesizing PAN membranes and PAN/ACB nanofiber composites using the electrospinning method. First, 0.16 g of PAN was dissolved into 1.84 N-Dimethylformamide (DMF) using a magnetic stirrer at 800 rpm at 80° for 1 h—this sample was called esPAN8. This was followed by the addition of 0.050 wt%, 0.065 wt%, and 0.075 wt% ACB into the PAN in DMF solution, which was stirred at 1000 rpm at room temperature for 2 h. Next, the PAN-ACB solution was sonicated for 1 h and stirred again using a magnetic stirrer at a speed of 1000 rpm at room temperature for 24 h. The final solution was deposited on an aluminum foil substrate using the electrospinning method with a voltage of 10 kV. Next, it was put in an oven at 75 °C for 5 h. All samples were characterized using SEM to determine morphology, fiber diameter, and membrane porosity. Sample codes are shown in Table 1.

### 2.2. Preparation of PAN-Based Electrolyte Membranes and PAN/ACB Nanofiber Composites

The nanofiber PAN and PAN/ACB membranes were dripped with a liquid electrolyte (mosalyte, TDE-250) until the nanofiber membrane had absorbed the electrolyte solution. Once cleaned, the solution that was left on the membrane surface and electrolyte membrane were obtained. Figure 2 shows a schematic of the electrolytic membrane preparation process based on PAN and PAN/ACB nanofiber membranes.

### 2.3. DSSC Synthesis and Fabrication

The DSSC photoanode was prepared by washing the FTO substrate first before depositing the blocking layer solution on the substrate using the spin coating method at 3000 rpm for 1 min. After that, it was heated gradually at 100 °C, 300 °C, and 500 °C for 15, 15, and 30 min, respectively. The TiO_2_ paste was deposited on top of the blocking layer using a screen-printing method and heated gradually. The samples were post-treated by immersing them in 20 mL 2-propanol and 150 µL titanium (IV) (triethanolamine) isopropoxide solution at a temperature of 75 °C for 30 min. They were then cleaned with ethanol pa and dried at 500 °C for 30 min; after that, the samples were immersed in dye N719 solution in a dark room for 17 h. Then, photoanodes were assembled in the order of photoanode/dye N179/electrolyte membrane (mosalyte, TDE-250)/Pt.

### 2.4. Characterization

The functional groups of the PAN membrane and PAN/ACB nanofiber composites were characterized by the Fourier Transform Infra-Red (FTIR) characterization of the Shimadzu IR-Prestige 21 type. The surface morphology of the nanofiber membranes was observed by the Scanning Electron Microscopy (SEM) characterization, which could then be analyzed for fiber diameter and nanofiber membrane porosity. The type of SEM used was SEM (FEI Inspect-S50 type, Malang, Indonesia). Electrolyte absorption on PAN membranes and PAN/AC nanofiber composites was carried out by taking the membranes of 1 × 1 cm. Then, the membrane was peeled off from the substrate and immersed in 50 µL for 1 h at room temperature. The photovoltaic characteristics of the DSSC were characterized by an *I-Vmeter* (Keithley 6517B, Malang, Indonesia) and solar simulator with a 150 W Xe Arc light source (LS 150, Azet Technologies, Malang, Indonesia). The active area was 0.5 × 0.5 cm with a light intensity of 100 mW/cm^2^ (1 sun). As a result, short circuit current density (*Jsc*), open circuit voltage (*Voc*), and power conversion efficiency (PCE) values were determined. The electrochemical impedance was characterized by electrochemical impedance spectroscopy (EIS) (Gamry reference 3000, Surabaya, Indonesia) under a light intensity of 100 mW/cm^2^.

## 3. Results

### 3.1. FT-IR (Fourier Transform Infra-Red)

The functional groups of the PAN and PAN/ACB nanofiber membrane composites were characterized using the FTIR spectrum. Figure 3 illustrates the wavenumber range of the IR spectrum, which ranges from 4000–500 cm^−1^.

### 3.2. SEM Image

The surface morphology and diameter of PAN membranes and PAN/ACB nanofiber composites were characterized using Scanning Electron Microscopy (SEM). Figure 4 is a SEM image of the PAN membrane and the PAN/ACB composite, which was observed at a magnification of 50,000 times. Figure 4a shows the result of an SEM image of a pure PAN nanofiber membrane synthesized with a PAN concentration of 8 wt%. The PAN nanofibers illustrate that the obtained fiber was smooth and uniform without beads. Meanwhile, the morphology of the PAN/ACB nanofiber composite membrane is shown in Figure 4b–d.

### 3.3. Electrolyte Absorption

Electrolyte absorption on the PAN membrane and the PAN/ACB composite was carried out by immersing the membrane in an electrolyte solution. This was followed by calculating the mass before and after the immersion processes, denoted as $W_{0}$ and $W_{i}$; the data are shown in Table 2.

### 3.4. J–V Curve and Electrochemical Impedance Specctroscopy

The characteristics of the *J–V* of the DSSC based on the PAN membrane and PAN/ACB nanofiber composite with an L-DSSC are shown in Figure 5. Table 3 illustrates the efficiency of the DSSC based on the PAN membrane (esPAN8) and PAN/ACB composites (esPACB1, esPACB2, and esPACB3). The increase in efficiency from 1.14% to 1.92% was due to the rise in membrane porosity [22]. In this research, the highest efficiency of 1.92% occurred in the esPACB1.

Furthermore, using electrochemical impedance spectroscopy, internal resistance and electron transport kinetics from the DSSC can be observed. Figure 6 is a Nyquist plot from the DSSC electrochemical impedance spectrum for all samples measured under the solar simulator with an intensity of 100 mW/cm^2^.

## 4. Discussion

### 4.1. PAN Membrane Function Groups and PAN/ACB Nanofiber Composite

The PAN molecule consists of methyl (CH_3_) and nitrile (C≡N) groups, which form several new compounds such as ketones, aldehydes, and carboxylic acids during the oxidation process [23]. Figure 3a shows the presence of several absorptions with high intensity at wave number 2243.21 due to the stretching of the C≡N nitrile group, thereby indicating the presence of acrylonitrile [23]. The high intensity indicated by wavenumber 1454.33 is a C-H aliphatic group (CH, CH2, CH3) formed due to the stretching of C=O [21,22,23,24,25]. Another low-intensity aliphatic group also appeared at wavenumbers 1247.94, 3626.17 and 1367.53, indicating stretching (CO), (CH), and (OH) bending [23,24,26,27,28,29]. The absorption that appears at wavenumbers 1730.15 and 1454.33 is related to the carboxylate function of the PAN group [28,29,32].

Figure 3b–d shows the FTIR spectra of the PAN membrane composited with ACB nanoparticles. It shows that the addition of ACB shows that no new peaks appeared, and there was only a difference in the absorption intensity in several numbers. These results showed that the C and N intermolecular interactions in PAN as a matrix and with ACB as a filler do not exist, so ACB nanoparticles are embedded in the PAN fiber as a filler [33]. The characteristics for each peak in the FTIR spectra of the PAN and PAN/ACB composites are shown in Table 4.

### 4.2. Diameter and Porosity of PAN Membranes and PAN/ACB Nanofiber Composite

The resulting fiber contains beads at several fiber points due to the addition of filler in the form of ACB with different masses, which causes a decrease in viscosity and an increase in conductivity [35]. Fiber diameter was obtained from SEM images and manually analyzed using ImageJ software. Based on Figure 4a, the diameter distribution shows that the average fiber diameter on the PAN nanofiber membrane is 206.1 ± 3.9 nm. Meanwhile, Figure 4b–d show that the average diameter of PAN membrane fibers that have been added with ACB nanoparticles varies, as shown in Table 5. Figure 7 shows the distribution of fiber diameters in a histogram.

The porosity of the PAN membranes and PAN/ACB nanofiber composites are represented in the surface porosity of the SEM images. Based on Table 6, the highest porosity of the PAN nanofiber membrane with a PAN concentration of 8 wt% is 65.76%. Meanwhile, the percentage of porosity of the PAN nanofiber membrane that had been composited with ACB ranges from 73.43 to 76.11%. In another study, Lee et al. reported the porosity of pure PAN membranes with a concentration of 10 wt%, which was 54% [36]. Meanwhile, for PAN membranes that have been composited with other nanoparticles, the percentage of porosity ranges from 50 to 52% [36]. A high porosity increases the surface area of the pores. Hence, the membrane is able to facilitate the absorption and retention of more liquid electrolytes to form a quasi-solid state with an ionic conductivity close to the appropriate liquid electrolyte while maintaining the superiority of the quasi-solid state [37,38]. With cross-linking on the nanofiber membrane that connects the nano fiber network, as seen from Figure 4, the increase of mechanical strength and stability tends to obtain good efficiency for DSSC cells effectively [16].

### 4.3. Electrolyte Absorption of PAN Membranes and PAN/ACB Nanofiber Composite

Electrolyte absorption from nanofiber membranes depends on the percentage of porosity and the pore structure of the membrane [8]. Electrolyte absorption on the PAN membrane and the PAN/ACB composite is carried out by immersing the membrane in an electrolyte solution. This is followed by calculating the mass before and after the immersion processes using Equation (1) [38], where $W_{0}$ and $W_{i}$ denote the mass of the membrane before and after immersion [38]:(1)$$Absorption\left(\%\right)=\frac{W_{i}-W_{0}}{W_{0}}\times 100\%$$

The percentage of electrolyte absorption on PAN membranes and PAN/ACB nanofiber composites are different due to the different porosity values of each nano-fiber membrane and the ability to absorb greater electrolytes [8]. The highest electrolyte absorption was shown by the PAN/ACB composite membrane with the addition of ACB of 0.0013 and 0.0015 g, which were 2050 and 2600%, respectively, as shown in Table 7. This result is known to be higher when compared to the study by Aydin et al., where a PAN membrane composited with hexagonal boron nitride showed the highest electrolyte absorption percentage value of 1790% [33] due to the high porosity of the PAN/ACB membrane, which was 76.11%. Hence, the absorption of the electrolyte becomes maximized [38].

### 4.4. DSSC Performance

Table 1 shows the photovoltaic parameters of the DSSC based on PAN membrane and PAN/ACB nanofiber composites as a medium for the electrolyte, including open-circuit voltage (*Voc*), short circuit current density (*Jsc*), fill factor (*FF*), and efficiency (*η*) [39]. Different effects can either impair or enhance the ionic conductivity, which later effects the performance of the PAN/ACB DSSC. This may account for the observed differences in the behavior of identical systems when nanoparticles are added.

The highest Jsc was shown by the esPACB1 membrane-based DSSC, which was 9.19 mA/cm^2^, corresponding to high efficiency. Meanwhile, the lowest *Jsc* was shown by liquid electrolyte-based DSSC cells—5.05 mA/cm^2^. The highest *J_SC_* observed depends on the transport of higher charge in both interfaces. Nanofiber electrolyte-based cells indicate medium charge transportation resistance. The tendency of variations in a large amount of charge transport resistor and the interface is correlated with *J_SC_* variations.

In most cases, liquid electrolyte-based cells show a short circuit current density, but the practical application is limited due to the problem of stability associated with the volatility [40]. Therefore, Nanofiber PAN-ACB-based solar cells show the efficiency of energy conversion that provides a better compromise for DSSC practical applications. The charge factor in the cells has the same value due to the same photoanode, which is based on TiO_2_. *V_OC_* is defined as the difference between the Fermi energy level of the TiO_2_ semiconductor and the electrolyte redox level under an open circuit [39]. A good *V_OC_* indicates a better photovoltaic performance. This research showed a high *V_OC_* of 0.63 V with an efficiency of 1.92%. This result is known to be higher when compared to another polymer electrolyte, namely lignin-based membranes that showed a DSSC cell efficiency of 1.13%.

Furthermore, the photo-electrochemical processes, including the ionic processes (diffusion and recombination) that occur in DSSC, have been investigated through EIS characterization. All DSSC Nyquist plots show almost the same semicircle spectrum, forming two semi-circles. This graph is then fitted using an EIS analyzer with a modified Randels–Ershler cell series, as shown in Figure 8. *R*_1_, *R*_2_, and *Ws* are ohmic resistance, charge transfer resistance, and Warburg impedance with finite length. *R*_1_ is known as series resistance. Its value decreases with each variation, causing the spectrum to shift to the left. This series resistance is associated with the diffusion of TiO_2_ shown in the first semicircle. Next is *R*_2_, where the resistance value is related to the diffusion of electrons and is shown in the second semi-circle where each variation looks bigger. Furthermore, this value can be used to determine ionic conductivity through Equation (2).
(2)$$\sigma \left(S\cdot {\mathrm{cm}}^{-1}\right)=\frac{l(\mathrm{cm})}{R\left(\Omega \right)A({\mathrm{cm}}^{2})}$$
where ‘*l*’ and *A* are the thickness (6.5 µm) and area (0.25 cm^2^) of the membrane, while *R* is the measurement result from the EIS at the electrolyte/TiO_2_ interface. The conventional double-layer capacitance is replaced by constant-phase elements (CPE) because the capacitance caused by the double-layer charging is distributed along the pores in the porous electrode. The fitting data results are presented in Table 8. The electrolyte membrane with the best ionic conductivity belongs to esPACB1, 1.71 × 10^−4^ S·cm^−1^. The excellent porosity and high ACB content also help to achieve high ionic conductivity. The diffusion of charge carriers enhances the ion exchange reactions due to the formation of electron pathways, increasing conversion efficiency.

The kinetics of charge transport in the Bode phase diagram can be used for electron life analysis in DSSC. Electron life is determined from the peak in the middle frequency range [41] through Equation (3) and presented in Table 9.
(3)$$\tau_{r}=\frac{1}{2\pi f_{max}}$$

It can be seen in Table 8 that the shortest electron lifetime (*τ_r_*) is shown for cells with esPACB1 membrane-based electrolytes, and the highest for cells with esPACB3 membrane-based electrolytes. This indicates that excited electrons in esPACB1 membrane-based electrolyte cells have a faster time to recombine with I_3_^−^ than esPACB3 membrane-based cells [41].

In this study, photoanodes were made using TiO_2_ films and assembled using electrolytes based on PAN and PAN/ACB nanofiber membranes. This study compared cell performances with various types of electrolyte media, such as DSSC using (i) liquid electrolyte, (ii) PAN nanofiber membrane electrolyte, and (iii) PAN/ACB nanofiber membrane electrolyte with different wt% ACB. Cells made using PAN/ACB nanofiber membranes, specifically esPACB1, can increase *J_SC_*∼82% from 5.05 to 9.19 mA/cm^2^ due to the short electron lifetime, which is 0.003 ms. Therefore, this cell can achieve a high conversion efficiency of 1.92%. This result turned out to be better than cells made using the liquid electrolytes.

## 5. Conclusions

In conclusion, PAN and PAN/ACB nanofiber composites membranes were successfully synthesized by the electrospinning method. The SEM image of the PAN nanofiber membranes showed smooth fibers without beads in a uniform fiber diameter at an average of (206.1 ± 3.9) nm. In contrast, the SEM image of the PAN/ACB composite membranes showed fibers and beads at several points in diameters of 172.9 to 284 nm. The FTIR results also confirmed this. The ACB nanoparticles were successfully embedded as fillers in the fiber. SEM imaging identified a porosity of 73.43% in the PAN/ACB composite membrane of esPACB1 with an electrolyte absorption of 1850% had the highest efficiency—1.92%. This is indicated as the optimum parameter that affects the membrane’s ability to absorb electrolytes and increase the ionic conductivity of the electrolyte. Thus, the DSSC cell performs with good efficiency.

## Figures and Tables

**Figure 1 micromachines-14-00394-f001:**
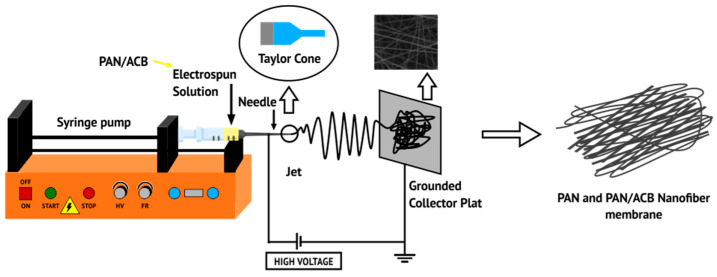
Schematic diagram illustrating the electrospinning process.

**Figure 2 micromachines-14-00394-f002:**
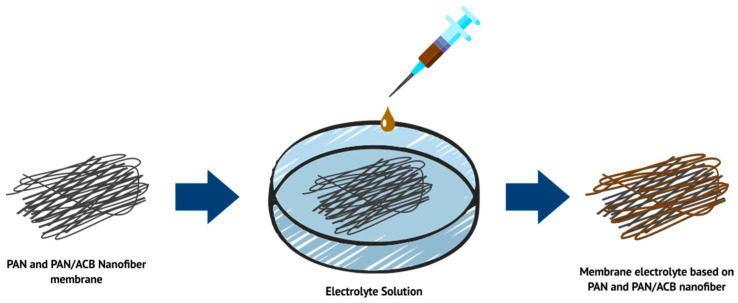
Schematic of the electrolytic membrane preparation process based on PAN nanofiber membranes and PAN/ACB composites.

**Figure 3 micromachines-14-00394-f003:**
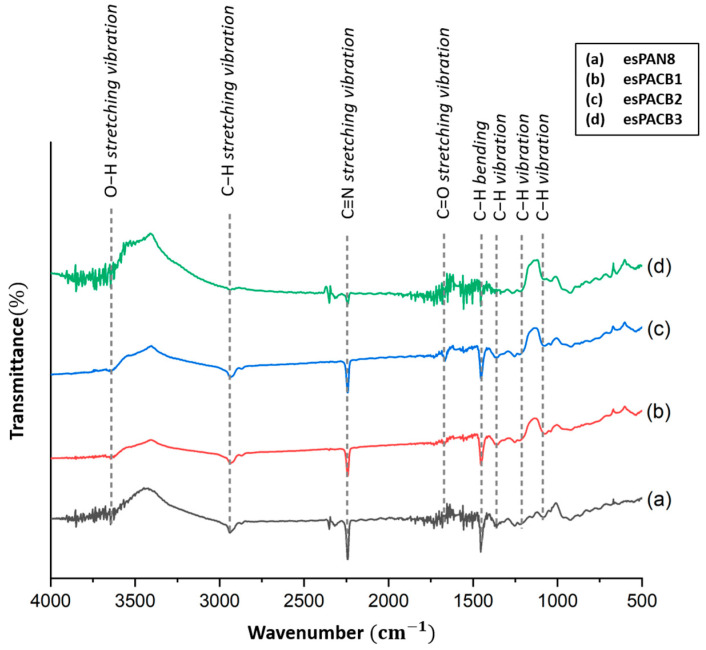
FTIR spectra of PAN and PAN/ACB nanofiber membrane composites.

**Figure 4 micromachines-14-00394-f004:**
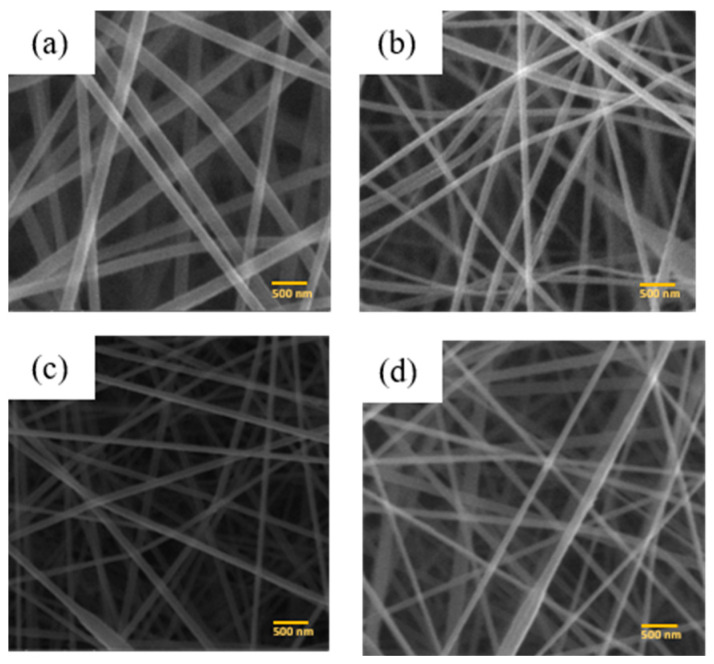
Surface morphology of PAN and PAN/ACB nanofiber composites (**a**) esPAN8, (**b**) esPACB1, (**c**) esPACB2 and (**d**) esPACB3.

**Figure 5 micromachines-14-00394-f005:**
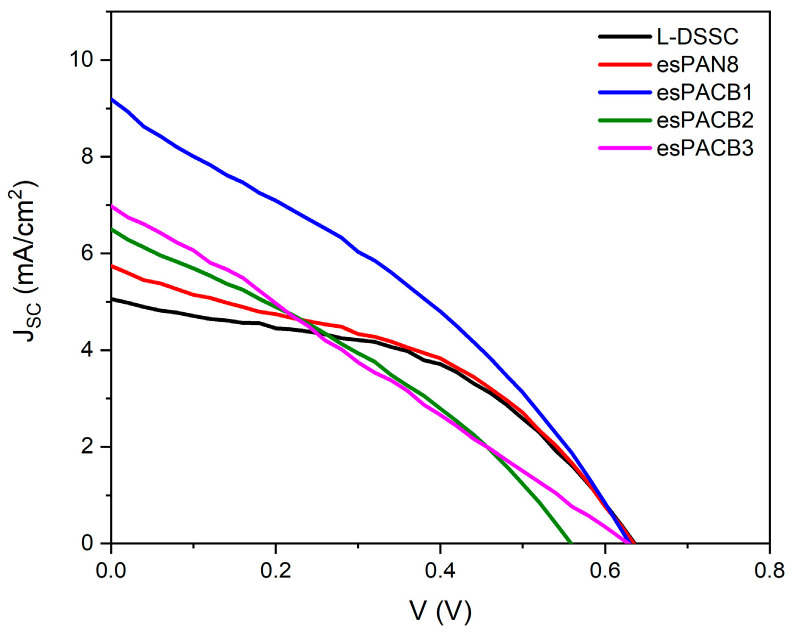
*J–V* curve of PAN membrane-based DSSC and PAN/ACB nanofiber composites.

**Figure 6 micromachines-14-00394-f006:**
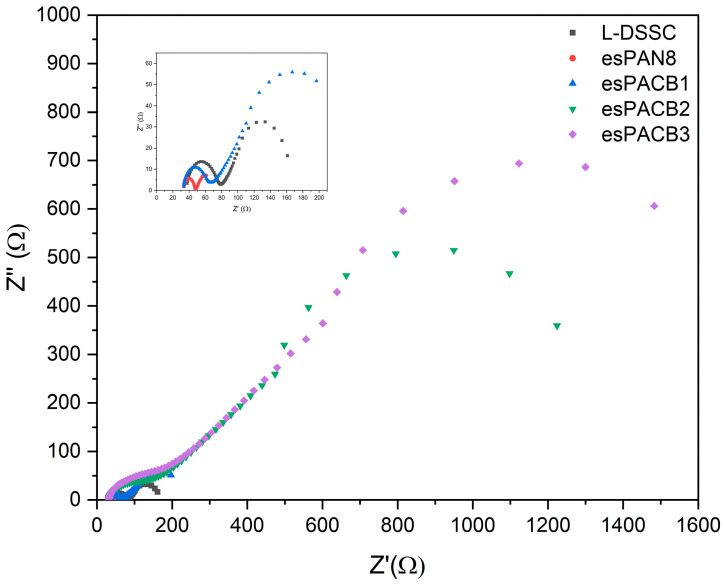
Nyquist plot of PAN membrane-based DSSC and PAN/ACB nanofiber composites.

**Figure 7 micromachines-14-00394-f007:**
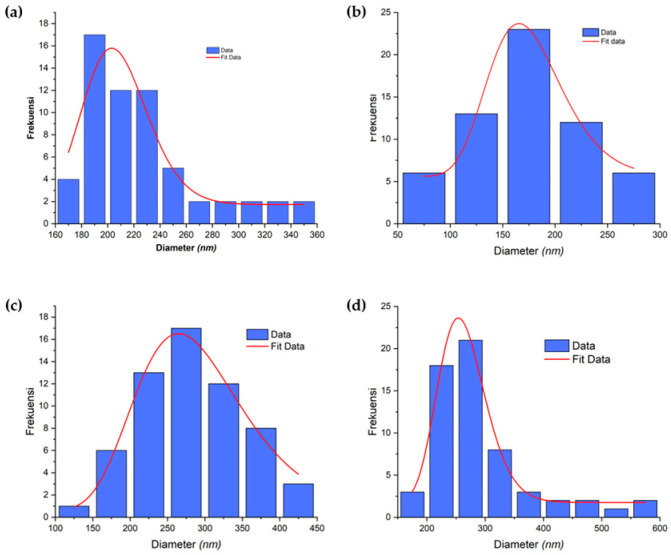
Diameter distribution of PAN membranes and PAN/ACB nanofiber composites and fiber diameters (**a**) esPAN8, (**b**) esPACB1, (**c**) esPACB2 and (**d**) esPACB3.

**Figure 8 micromachines-14-00394-f008:**
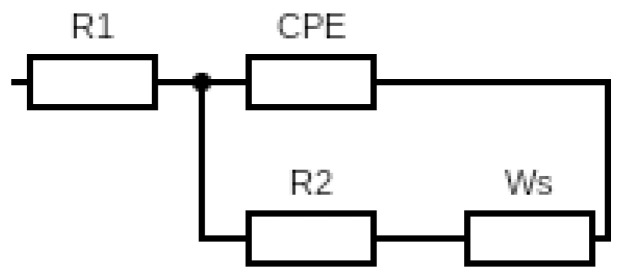
Equivalent circuit of PAN membrane-based DSSC photovoltaic and PAN/ACB nanofiber composites.

**Table 1 micromachines-14-00394-t001:** Sample code of PAN and PAN/ACB nanofiber composites.

Sample Code	ACB Addition (wt%)
esPAN8	0
esPACB1	0.050
esPACB2	0.065
esPACB3	0.075

**Table 2 micromachines-14-00394-t002:** Mass before and after immersion process of PAN and PAN/ACB nanofiber composites.

Sample Code	$W_{0}$ (g)	$W_{i}$ (g)
esPAN8	0.0003	0.0027
esPACB1	0.0002	0.0039
esPACB2	0.0002	0.0043
esPACB3	0.0002	0.0054

**Table 3 micromachines-14-00394-t003:** Parameters of PAN membrane-based DSSC photovoltaic and PAN/ACB nanofiber composites.

Sample Code	*I_SC_* (mA)	*J_sc_* (mA/cm^2^)	*I_max_* (mA)	*V_max_* (V)	*Voc* (V)	*FF*	*η* (%)
L-DSSC	1.24	5.05	0.88	0.42	0.63	0.46	1.48
esPAN8	1.43	5.74	0.95	0.40	0.63	0.42	1.53
esPACB1	2.29	9.19	1.26	0.38	0.63	0.33	1.92
esPACB2	1.62	6.49	0.93	0.32	0.55	0.33	1.20
esPACB3	1.74	6.97	0.84	0.34	0.63	0.26	1.14

**Table 4 micromachines-14-00394-t004:** Comparison of PAN membranes and PAN/ACB nanofiber composites FTIR spectra based on Figure 3.

Wavenumber (cm^−1^)			Bonding Characteristics
Nanofiber PAN	Nanofiber PAN-ACB	Reference	
1078.21	1080.14	~1190–1390	Methylene (C−H) group [29]
1247.94	1255.66	~1220–1270	Aliphatic C−H (CH, CH_2_, CH_3_) groups vibration [24]
1367.53	1363.67	~1350–1380	Aliphatic C−H (CH, CH_2_, CH_3_) groups vibration [24]
1454.33	1450.47	~1450–1460	Aliphatic C−H (CH, CH_2_, CH_3_) groups vibration [24,26,34]
1730.15	-	1736	C=O stretching vibration [28,29]
2243.21	2241.28	~2241–2265	C≡N nitrile group stretching vibration [25,26,27]
2929.87	2939.51	~2925–2942	Stretching vibration C−H (CH, CH_2_, CH_3_) groups [25,26,30,31]
3626.17	3635.81	3477	O−H stretching vibration [25,31]

**Table 5 micromachines-14-00394-t005:** The average diameter of PAN membranes and PAN/ACB nanofiber composites.

Sample Code	Diameter (nm)
esPAN8	206.1 ± 3.9
esPACB1	172.9 ± 2.2
esPACB2	284.0 ± 5.2
esPACB3	260.3 ± 0.9

**Table 6 micromachines-14-00394-t006:** Percentage porosity of PAN membranes and PAN/ACB nanofiber composites.

Sample Code	∅(%)
esPAN8	65.76
esPACB1	73.43
esPACB2	75.84
esPACB3	76.11

**Table 7 micromachines-14-00394-t007:** Percentage of electrolyte absorption in PAN membranes and PAN/ACB nanofiber composites.

Sample Code	Absorption (%)
esPAN8	800
esPACB1	1850
esPACB2	2050
esPACB3	2600

**Table 8 micromachines-14-00394-t008:** EIS Parameter of PAN membrane-based DSSC photovoltaic and PAN/ACB nanofiber composites.

Sample Code	*R*_1_ (Ω)	*R*_2_ (Ω)	*CPE* (S·s^a^ × 10^−5^)	*W_s_* (Ω)	*σ* (S·cm^−1^)
L-DSSC	35.001	44.014	1.626	42.944	5.91 × 10^−5^
esPAN8	32.748	15.229	1.248	7.4037	6.61 × 10^−5^
esPACB1	30.587	39.313	4.544	66.935	1.71 × 10^−4^
esPACB2	28.65	143.95	5.231	557.66	1.81 × 10^−5^
esPACB3	28.422	183.2	4.673	726.04	1.42 × 10^−5^

**Table 9 micromachines-14-00394-t009:** Frequency and electron lifetime of PAN membrane-based DSSC photovoltaic and PAN/ACB nanofiber composites.

Sample Code	Frequency (Hz)	Electron Lifetime (ms)
L-DSSC	3981.07	0.039
esPAN8	7943.28	0.020
esPACB1	39810.7	0.003
esPACB2	631.00	0.252
esPACB3	398.10	0.399

## Data Availability

Data are contained within the article.

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
