# Peer review of "The Effect of Acetylene Carbon Black (ACB) Loaded on Polyacrylonitrile (PAN) Nanofiber Membrane Electrolyte for DSSC Applications"

_micromachines, 2023, doi:10.3390/mi14020394_

Round 1

Reviewer 1 Report

The manuscript entitled "The Effect of Acetylene Carbon Black (ACB) Loaded on Polyacrylonitrile (PAN) Nanofiber Membrane Electrolyte for DSSC Applications" has a novelty work. The authors prepared PAN-ACB membrane composite in this work by directly electrospinning nanofibers onto an aluminum foil substrate. Influence of dif- 66 different ACB masses incorporated onto membrane composite towards the electrolytes absorb- 67 ance ability and ionic conductivity is discussed in detail for its potential application in DSSC. The introduction and the aim of the work are obvious. The experimental work is explained well. Also, the results and conclusions are looks good, and the interpretation is very clear. So, I recommended that the paper is acceptable in its present form.

Author Response

We would like to thanks the reviewer for providing the time to read carefully and thouroughly of this manuscript, the thoughtful comments, and the recommendation for accepting this paper.

Reviewer 2 Report

The manuscript entitled “The Effect of Acetylene Carbon Black (ACB) Loaded on Polyacrylonitrile (PAN) Nanofiber Membrane Electrolyte for DSSC  Applications” has been submitted by the authors. Some issues to be addressed will improve the quality of the manuscript. Therefore, I recommend this work could be published after the major revision

1.       The author should write down the novelty of this paper.

2.       The English composition requires many improvements. The authors should proofread the manuscript carefully to minimize grammatical errors.

3.       Check the format of the reference and correct all the errors.

4.       In the introduction, the author should like down the comparative study best on this study.

5.       The author should explain with adding acetylene carbon black, I am wounder it increases the dark current, please explain the concept.

6.       Why the FF of esPACB1 decrease, any reason behind that? 

Author Response

We would like to thank the reviewer for providing the time to reading carefully and thoroughly of this manuscript and for the thoughtful comments and valuable suggestions. The corresponding changes of the revised paper are summarized in our response below. Please see the attachment.

Reviewer 3 Report

The manuscript entitled “The Effect of Acetylene Carbon Black (ACB) Loaded on Polyacrylonitrile (PAN) Nanofiber Membrane Electrolyte for DSSC Applications” has been submitted by the authors. Some issues to be addressed will improve the quality of the manuscript. Therefore, I recommend this work could be published after the major revision

1.      The author should write the result value in abstract

2.      The introduction needs to be modified with some latest references.

3.      English language needs to be polished?

4.      Can author provide any experimental support to prove ionic conductivity?

5.      Author should give details of sample code when it’s come on first

6.      To calculate power conversion efficiency how many samples authors used?

Author Response

(The authors gave the same response as above.)

Round 2

Reviewer 2 Report

The author carefully answers all of the comments.  i recommended to accept it in present form

Reviewer 3 Report

The author solve all comments very carefully i recommended to accept in present form.